

# Growth of nucleation mode particles in the summertime Arctic: a case study

Megan D. Willis[1], Julia Burkart[1], Jennie L. Thomas[2], Franziska Köllner[3], Johannes Schneider[3], Heiko Bozem[4], Peter M. Hoor[4], Amir A. Aliabadi[5,a], Hannes Schulz[6], Andreas B. Herber[6], W. Richard Leaitch[5], and Jonathan P.D. Abbatt[1]

[1]University of Toronto, Department of Chemistry, Toronto, Ontario, Canada
[2]LATMOS/IPSL, UPMC Univ. Paris 06 Sorbonne Universités, UVSQ, CNRS, Paris, France
[3]Max Planck Institute for Chemistry, Particle Chemistry Department, Mainz, Germany
[4]Johannes Gutenberg University of Mainz, Institute for Atmospheric Physics, Mainz, Germany
[5]Environment and Climate Change Canada, Toronto, Ontario, Canada
[6]Alfred Wegener Institute Helmholtz-Center for Polar and Marine Research Bremerhaven, Bremerhaven, Germany
[a]Now at: Massachusetts Institute of Technology, Department of Architecture, Cambridge, USA

*Correspondence to:* Megan D. Willis (megan.willis@mail.utoronto.ca)

**Abstract.** The summertime Arctic lower troposphere is a relatively pristine, background aerosol environment dominated by nucleation and Aitken mode particles. Understanding the mechanisms that control the formation and growth of aerosol is crucial for our ability to predict cloud properties, and therefore radiative balance and climate. We present an analysis of an aerosol growth event observed

in the Canadian Arctic Archipelago during summer as part of the NETCARE project. Under stable and clean atmospheric conditions, with low inversion heights, carbon monoxide less than 80 $ppb_v$ and black carbon less than 5 $ng\,m^{-3}$, we observe growth of small particles, <20 nm in diameter, into sizes above 50 nm. Aerosol growth was correlated with the presence of organic species, trimethylamine and methanesulfonic acid (MSA) in particles ∼80 nm and larger, where the organ-

ics are similar to those previously observed in marine settings. MSA-to-sulfate ratios as high as 0.15 were observed during aerosol growth, suggesting an important marine influence. The organic-rich aerosol contributes significantly to particles active as cloud condensation nuclei (CCN, supersaturation = 0.6%), which are elevated in concentration during aerosol growth above background levels of ∼100 $cm^{-3}$ to ∼220 $cm^{-3}$ . Results from this case study highlight the potential importance of sec-

ondary organic aerosol formation and its role in growing nucleation mode aerosol into CCN-active sizes in this remote marine environment.

## 1 Introduction

In the warming Arctic (Jeffries and Richter-Menge, 2012), decreasing sea ice extent (Lindsay et al., 2009) is likely to increase the oceanic influence on atmospheric composition. This change in exposed





ocean area will have implications on aerosol concentrations and composition, and therefore on cloud
properties (Browse et al., 2014) and precipitation (Kopec et al., 2016). Aerosol-cloud-climate inter-
actions are unique in Arctic regions due to the high surface albedo, the seasonal cycle in aerosol
loading and properties, the strong static stability in the lower troposphere (Aliabadi et al., 2016b)
and the dependence of cloud infrared emissivity on droplet size and aerosol characteristics (Curry,
25 1995).

Pristine, background aerosol conditions prevail in the summertime Arctic boundary layer. A pro-
nounced seasonal cycle characterizes Arctic aerosol (Engvall et al., 2008; Sharma et al., 2013;
Tunved et al., 2013; Croft et al., 2015; Nguyen et al., 2016), with strong anthropogenic contributions
to "Arctic Haze" in winter and spring (Law and Stohl, 2007; Quinn et al., 2007), and more regional
influences in the cleaner summer months, especially in the lower troposphere (Leaitch et al., 2013;
Heintzenberg et al., 2015). Beginning in late spring, efficient wet removal of aerosol and less effi-
cient transport from lower latitudes come together to suppress the condensation sink (Stohl, 2006;
Engvall et al., 2008) and allow nucleation and Aitken mode particles to dominate the size distribution
(Engvall et al., 2008; Heintzenberg and Leck, 2012; Croft et al., 2015). Under these clean conditions
cloud condensation nuclei (CCN) and cloud droplet number concentrations can be exceptionally low
(Mauritsen et al., 2011; Leaitch et al., 2016), making summertime liquid clouds very sensitive to the
formation of new particles and their growth into CCN sizes. Since Arctic clouds are an important de-
terminant of the local surface energy balance (e.g., Intrieri et al., 2002; Lubin and Vogelmann, 2006)
and have the ability to influence the thickness, freezing and melting of sea ice (Kay and Gettelman,
2009; Tjernström et al., 2015), a predictive understanding of the sources and processes controlling
CCN-active aerosol is a crucial aspect of understanding the Arctic climate.

While transport of pollutants from lower latitudes does occur in Arctic summer, especially in the
middle and upper troposphere, efficient scavenging during transport and within Arctic regions results
in an important contribution from regional sources near the surface at this time of year (e.g., Stohl,
2006; Garrett et al., 2011; Croft et al., 2015). In the absence of significant transported aerosol, several
different processes can contribute to aerosol formation, including the emission of primary particles
from the ocean surface, along with formation of new particles by nucleation and their subsequent
growth by condensation and coagulation.

The formation of new particles can be an important aerosol source in the summertime Arctic
(Leaitch et al., 2013; Croft et al., 2015). Through its oxidation to sulfuric acid and other prod-
ucts, dimethyl sulfide (DMS) plays an important role in the formation, and growth, of new particles
(Leaitch et al., 2013). In the Arctic and at mid-latitudes, uncertainties in the rates and mechanisms
of nucleation and growth are such that some studies are able to explain ambient observations with
standard parametrizations developed from measurements at more southerly locations (e.g., Chang
et al., 2011b), while others must invoke alternative mechanisms (e.g., Karl et al., 2012). The role of
ammonia and amines in particle nucleation at mid-latitudes has become well established (Almeida



et al., 2013), and recent measurements suggest that local ammonia sources in the summer Arctic are sufficient to promote particle formation (Wentworth et al., 2015; Giamarelou et al., 2016). Iodine oxides can make a significant contribution to new particle formation in marine and coastal envi-
ronments at mid-latitudes (e.g., O'Dowd and de Leeuw, 2007); these species may contribute to the formation and growth of small particles in Arctic regions, although their biotic and abiotic sources in ice-covered regions remain unclear (Mahajan et al., 2010; Allan et al., 2015). Organic condensible species also play a role in nucleation, and growth, of particles at mid-latitudes (e.g., Kulmala and Kerminen, 2008; Metzger et al., 2010; Ehn et al., 2014); however, no direct evidence for the role of
organic species in Arctic nucleation events exists to date.

The ejection of primary aerosol from the sea surface, through wave-breaking and bubble-bursting, is another source of aerosol across the size distribution (Ovadnevaite et al., 2014; Clarke et al., 2006; Nilsson et al., 2001). At mid-latitudes a large organic fraction, which originates from the enrichment of biologically-derived organic material at the sea surface, is frequently observed in marine aerosol
(Facchini et al., 2008b; Gantt and Meskhidze, 2013; Frossard et al., 2014; Quinn et al., 2015a; O'Dowd et al., 2015; Quinn et al., 2015b). This primary marine organic aerosol (OA) tends to be water-insoluble with chemical similarity to lipids (e.g., Rinaldi et al., 2010; Decesari et al., 2011), and has been demonstrated to have a source near the ocean surface (Ceburnis et al., 2008). Some similar observations have been made in Arctic regions (e.g., Narukawa et al., 2008; Orellana et al.,
2011; Fu et al., 2013; Karl et al., 2013; Fu et al., 2015). For example, Fu et al. (2013, 2015) have shown a dominance of primary saccharides and evidence for protein and humic-like substances in Arctic aerosol suggesting an important local or regional source of primary marine OA. The release of marine micro-gels via bubble-bursting in open leads has been proposed to contribute significantly to particles over the Arctic Ocean (e.g., Bigg and Leck, 2001; Orellana et al., 2011).

Particle growth through condensation of gas-phase species can also play a role in driving marine aerosol characteristics, making ambient marine OA a complex result of primary and secondary processes (e.g., Ceburnis et al., 2008; Facchini et al., 2008b; Rinaldi et al., 2010; Frossard et al., 2014). In contrast to primary marine OA, secondary marine OA is generally more water-soluble and is composed of more oxygenated compounds (Rinaldi et al., 2010; Decesari et al., 2011). Precursors of
secondary marine OA include DMS and other biological volatile organic compounds (BVOCs), such as isoprene, monoterpenes and amines, which are produced by a variety of marine micro-organisms (Shaw et al., 2010; Gantt et al., 2009; Facchini et al., 2008a). However, in the absence of specific molecular tracers it can be very challenging to discern the relative contribution of primary and secondary processes to ambient marine organic aerosol (e.g., O'Dowd et al., 2015). At mid-latitudes,
direct and indirect measurements of Aitken mode particle composition have demonstrated the role of secondary organic species in the growth of small particles (Vaattovaara et al., 2006; Bzdek et al., 2014; Lawler et al., 2014). Significant fractions of alkylamines, dicarboxylic acids, methansulfonic acid, oxalic acid, alcohols and other organic acids have been observed in marine aerosol, suggest-



ing contributions from secondary processes (e.g., Facchini et al., 2008a; Claeys et al., 2010; Rinaldi
et al., 2010; Dall'Osto et al., 2012; Frossard et al., 2014). In Arctic regions, the detection of specific
molecular tracers for isoprene, terpene and fatty acid oxidation have indicated a contribution of sec-
ondary processes to summertime organic aerosol (Fu et al., 2009; Kawamura et al., 2012; Fu et al.,
2013; Hansen et al., 2014).

Our understanding of summertime Arctic aerosol remains incomplete, in part due to a scarcity
of observations focusing on the influence of local and regional sources on aerosol chemical and
physical properties. In this case study we focus on observations of a new particle formation and
growth event made during the NETCARE summer aircraft campaign in July 2014, near Resolute
Bay, Nunavut, Canada, in a general time period and location that was shown to have high biological
activity in the surface ocean (Gosselin et al., 2015; Mungall et al., 2015). We use these observations
to explore the composition and formation processes of particles contributing to cloud condensation
nuclei in the Canadian Arctic Archipelago during summer.

## 2 Methods

### 2.1 Measurement platform and inlets

As part of the NETCARE project (Network on Climate and Aerosols: Addressing Key Uncertainties
in Remote Canadian Environments, http://www.netcare-project.ca), measurements of aerosol physi-
cal and chemical properties, trace gases and meteorological parameters were made aboard the Alfred
Wegener Institute (AWI) Polar 6 aircraft; a DC-3 aircraft converted to a Basler BT-67 (Herber et al.,
2008). Measurements aboard Polar 6 took place from July 4 – 21, 2014, based in Resolute Bay,
Nunavut (74° 41' N, 94° 52' W). The survey speed was maintained at approximately 120 knots ($\sim$
222 $\mathrm{km\,h^{-1}}$) for measurement flights, with ascent and descent rates of 150 $\mathrm{m\,min^{-1}}$ for vertical
profiles.

The main aerosol inlet was located on the starboard side of the fuselage ahead of the engines.
Based upon a total flow drawn to instruments of 35 $\mathrm{L\,min^{-1}}$ and a measured flow at the exhaust
of the sampling line of 20 $\mathrm{L\,min^{-1}}$, the total flow through the shrouded inlet diffuser was nearly
isokinetic at 55 $\mathrm{L\,min^{-1}}$. Aerosol flowed into the cabin through a stainless steel manifold (outer
diameter = 1.9 cm) and was directed to the various particle instruments through stainless steel lines
that branched from the main inlet at angles less than 90 degrees. Aerosol was not dried prior to
sampling; however, the temperature in the inlet line was approximately 10 – 15 °C warmer than the
ambient temperature so that the relative humidity (RH) decreased significantly as the aerosol entered
the sampling line. Exhaust from the main aerosol inlet flowed freely into the back of the cabin to
keep the inlet from being over-pressured. Therefore, the total flow through the main aerosol inlet was
dictated by the true airspeed (TAS). With the survey air speed noted above, transmission efficiency
of aerosol through the main inlet was near unity for particles 20 nm to $\sim$1 μm in diameter.



Trace gases (CO, $CO_2$ and $H_2O$) were sampled through a second inlet consisting of a 0.40 cm

(outer diameter) Teflon line, with a continuously measured sample of flow of $\sim 12$ L min$^{-1}$. The

trace gas inlet used the forward motion of the aircraft to push ambient air into the line in combination

with a rear-facing 0.95 cm Teflon exhaust line that lowered the pressure in the sampling line.

## 2.2   State parameters and winds

State parameters and meteorological conditions were measured with an AIMMS-20, manufactured

by Aventech Research Inc. (Barrie, Ontario, Canada http://aventech.com/products/aimms20.html).

The AIMMS-20 consists of three modules: (1) an Air Data Probe, which measures temperature

and the three-dimensional aircraft-relative flow vector (TAS, angle-of-attack and side-slip) with a

three-dimensional accelerometer for measurement of turbulence; (2) an Inertial Measurement Unit,

which provides the aircraft angular rate and acceleration; (3) a Global Positioning System for aircraft

three-dimensional position and inertial velocity. Vertical and horizontal wind speeds are measured

with accuracies of 0.75 and 0.50 m s$^{-1}$, respectively. Accuracy and precision of the temperature

measurement are 0.30 and 0.10 °C, respectively.

## 2.3   Aerosol physical properties

Measurements of particle number concentrations, and size, were made aboard Polar 6 at a frequency

of 1 Hz, unless otherwise indicated. Number concentrations of particles greater than 5 nm in di-

ameter ($N_{>5}$) were measured with a TSI 3787 water-based ultra-fine condensation particle counter

(UCPC), sampling at a flow rate of 0.6 L min$^{-1}$. Aerosol number size distributions from 20 nm to

1 µm were acquired with two instruments: a Brechtel Manufacturing Incorporated (BMI) Scanning

Mobility System (SMS) coupled to a TSI 3010 Condensation Particle Counter (CPC) measured from

150 20 to 100 nm ($N_{20-100}$) with a 60-second time resolution, while a Droplet Measurement Technology

(DMT) Ultra High Sensitivity Aerosol Spectrometer (UHSAS) measured number size distributions

from 70 nm to 1 µm ($N_{>70}$) with a time resolution of 1 Hz. The SMS sampled at a flow rate of

1 L min$^{-1}$, with a dried ($\sim$20% RH) sheath flow of 6 L min$^{-1}$. The UHSAS uses light scattering

signals from a 1054 nm laser for particle detection and sizing on a single-particle basis (e.g., Cai

155 et al., 2008), with a sample flow rate of 55 cm$^3$ min$^{-1}$ from a bypass flow off the main aerosol inlet.

Characterization and calibration of the UCPC, SMS, UHSAS and CCNC are described in detail in

Leaitch et al. (2016). Agreement between the particle instruments was generally within a factor of

two.

Particle number concentrations from all instruments are reported at ambient pressure and tem-

160 perature. Values of $N_{>80}$, $N_{>100}$ and $N_{>200}$ were derived from UHSAS measurements. Number

concentrations from 5 to 20 nm ($N_{5-20}$) were estimated by subtracting the sum of the SMS total

number concentration ($N_{20-100}$) and UHSAS $N_{>100}$ from the total UCPC concentration (i.e., $N_{>5}$).

The number of particles greater than 50 nm ($N_{>50}$) was determined by the sum of the SMS number



from 50 to 100 nm ($N_{50-100}$) and the UHSAS $N_{>100}$. Here, we refer to $N_{5-20}$ as the nucleation
mode, $N_{20-100}$ as the Aitken mode and $N_{>100}$ and larger as the accumulation mode.

## 2.4 Cloud condensation nuclei concentrations

Cloud Condensation Nuclei (CCN) concentrations were measured using a DMT CCN counter (CCNC,
Model 100), sampling behind a DMT pressure controlled inlet at a reduced pressure of $\sim$650 hPa.
The effective supersaturation (for a nominal water supersaturation of 1%, at 650 hPa) was found to
be 0.6% (Leaitch et al., 2016), and was held constant throughout the study to allow more measure-
ment stability and the highest time resolution possible, and to examine the hygroscopicity of small
particles.

The effective aerosol hygroscopicity parameter ($\kappa$) was estimated according to Petters and Krei-
denweis (2007), using the average aerosol composition from the aerosol mass spectrometer (see
below) with ammonium sulfate and organic aerosol densities of 1770 $\mathrm{kg\,m^{-3}}$ and 1550 $\mathrm{kg\,m^{-3}}$,
respectively (e.g., Chang et al., 2010). Assuming a temperature of 298 K and the surface tension of
pure water the dry diameter for activation was calculated at the supersaturation of our CCN measure-
ments. The measured size distribution could then be integrated down to this dry diameter to produce
predicted CCN concentrations for comparison with measured values.

## 2.5 Trace gases

Carbon monoxide (CO) concentrations were measured with an Aerolaser ultra-fast carbon monoxide
monitor (model AL 5002), based on VUV-fluorimetry using excitation of CO at 150 nm. The instru-
ment was modified such that in-situ calibrations could be conducted in flight. CO concentrations are
used here as a relative indicator of aerosol influenced by pollution sources, such as anthropogenic or
biomass burning emissions.

Water vapour ($H_2O$) measurements were based on infrared absorption using a LI-7200 enclosed
$CO_2/H_2O$ Analyzer from LI-COR Biosciences GmbH. In-situ calibrations were performed during
flight at regular intervals (15 – 30 min) using a NIST traceable $CO_2$ standard with zero water va-
por concentration. The measurement uncertainty is $\pm$ 15 $\mathrm{ppm_v}$. $H_2O$ mixing ratios were used to
calculate relative humidity with pressure and temperature measured by the AIMMS-20.

## 2.6 Sub-micron aerosol composition

### 2.6.1 Single particle soot photometer

Concentrations of refractory black carbon (rBC) containing particles were measured with a DMT
single particle soot photometer (SP2) (described in Schwarz et al. (2006) and Gao et al. (2007)),
and are used as an indicator of pollution influences. The SP2 uses a continuous intra-cavity Nd:YAG
laser (1064 nm) to classify particles as either incandescent (rBC) or scattering (non-rBC), based on



the individual particle's interaction with the laser beam. The peak incandescence signal is linearly related to the rBC mass. The SP2 was calibrated with an Aquadag standard by selecting a narrow size distribution of particles with a differential mobility analyzer upstream of the SP2 (Schwarz

et al., 2006; Laborde et al., 2012). The detection efficiency of this SP2 (version D) drops off for particles smaller then 60 nm. A log-normal fit through the mass-size distribution indicates that the SP2 measured 92% of the total ambient rBC mass. Reported rBC values were thus scaled up by a factor of 1.08 to account for the fraction of rBC particles that were outside of the SP2 detection range. The SP2 sampled at 120 $cm^3 min^{-1}$, sharing a bypass line from the main aerosol inlet with

the UHSAS.

### 2.6.2 Aerosol mass spectrometer

Sub-micron aerosol composition was measured with an Aerodyne high-resolution time-of-flight aerosol mass spectrometer (HR-ToF-AMS), described in detail by DeCarlo et al. (2006). The HR-ToF-AMS deployed here was equipped with an infrared laser vaporization module similar to that of

the SP2 (DMT); however, measurements of refractory black carbon (rBC) are not relevant for the data presented here owing to extremely low rBC concentrations. The HR-ToF-AMS was operated in "V-mode" with a mass range of m/z $3 - 250$, alternating between ensemble mass spectrum (MS) mode for 20 s (two cycles of 5 (s) MS open and 5 (s) MS closed) and efficient particle time-of-flight (epToF) mode for 10 s. Filtered ambient air was sampled with the HR-ToF-AMS approximately

three times per flight, for a duration of at least five minutes, to account for contributions from air signals. Data were analysed using the Igor Pro-based analysis tool PIKA (v.1.16) and SQUIRREL (v.1.57) (Seuper, 2010).

The HR-ToF-AMS sampled behind a pressure-controlled inlet (PCI) system, similar to that described by Hayden et al. (2011), in order to remove variations in particle sizing, transmission and

air-beam signals (used to correct particle signals for variations in instrument sensitivity (Allan et al., 2003)) as a function of pressure in the aerodynamic lens (Bahreini et al., 2008; DeCarlo et al., 2008). The PCI system maintained a pressure of $6.19 \times 10^4$ Pa upstream of a 130 μm orifice in the HR-ToF-AMS inlet and downstream from a 200 μm orifice, such that the pressure in the aerodynamic lens was maintained at 173 Pa ($\sim$1.3 Torr) by variable pumping. In this configuration, the lens pressure

was adequately maintained up to an altitude of $\sim$3500 m. Characterization of particle transmission efficiency with and without the PCI was carried out before and after the study (Section 1.1 in the Supplement). Results demonstrated near 100% transmission of ammonium nitrate particles from $\sim$70 – 700 nm (mobility diameter) through the PCI, by comparison to transmission through the aerodynamic lens alone (Figure S1 in the Supplement). Note that the size range over which AMS

lens transmission is optimal can be very instrument dependent. HR-ToF-AMS particulate mass loadings are corrected to reflect ambient pressure, based on the AIMMS measured pressure and the PCI internal pressure.



Species comprising non-refractory particulate matter are measured with the HR-ToF-AMS, including sulfate, nitrate, ammonium and the sum of organic species, with an uncertainty of $\pm 30\%$

(Bahreini et al., 2009). The HR-ToF-AMS is capable of detecting other species, including methanesulfonic acid (Phinney et al., 2006; Zorn et al., 2008) and sea salt (Ovadnevaite et al., 2012).

The detection efficiency of sea salt containing particles is dependent on not only the ambient relative humidity, but also the temperature of the tungsten vaporizer (Ovadnevaite et al., 2012). The vaporizer temperature was calibrated with sodium nitrate particles, such that particle-time-of-flight

signals indicated efficient vaporization, and was operated at a temperature of $\sim 650\,^{\circ}\mathrm{C}$. HR-ToF-AMS signals for sea salt, in particular $NaCl^+$ (m/z 57.96), can be used to quantify sea salt mass loadings (e.g., Ovadnevaite et al., 2012); however, here we use the $NaCl^+$ signal only as a qualitative indication for the presence of sea salt.

After the method of Zorn et al. (2008), we determined the fragmentation pattern for methanesul-

fonic acid (MSA) under the operating conditions of our HR-ToF-AMS by utilizing the unique MSA fragment $CH_3SO_2^+$ (m/z 78.99), which was well resolved from organic fragments at the same nominal mass (i.e., $C_6H_7^+$ at m/z 79.05, Figure S2 in the Supplement). The default HR-ToF-AMS fragmentation table was modified to include MSA, such that contributions from MSA to peaks usually associated with organic species and sulfate were accounted for. The sensitivity of our HR-ToF-AMS

to MSA relative to nitrate ($RIE_{MSA}$) was determined to be $1.33 \pm 0.05$, which is similar to estimated values used in other studies (e.g., Zorn et al., 2008). The MSA calibration and fragmentation pattern are described in more detail in Section 1.1 of the Supplement.

Ammonium nitrate calibrations (with 300 nm particles) were carried out four times during the campaign (Jimenez et al., 2003), and air-beam corrections were referenced to the appropriate cal-

ibration in order to account for differences in instrument sensitivity between flights. The relative ionization efficiencies for sulfate and ammonium ($RIE_{SO_4}$ and $RIE_{NH_4}$) were $1.4\pm0.1$ and $3.7\pm0.3$. The default relative ionization efficiency for organic species (i.e., $RIE_{Org} = 1.4$) was used (Jimenez et al., 2003), which may lead to some larger uncertainty in the quantification of organic aerosol mass (Murphy, 2015). Elemental composition was calculated using the method presented in Cana-

garatna et al. (2015). Detection limits for sulfate, nitrate, ammonium, MSA and organics based on three-times the signal-to-noise of filter measurements in flight were 0.009, 0.008, 0.004, 0.005 and 0.08 $\mu g\,m^{-3}$, respectively. A composition dependent collection efficiency was applied to correct HR-ToF-AMS mass loadings for non-unity particle detection due to particle bounce on the tungsten vaporizer (Middlebrook et al., 2012).

**2.6.3 Aircraft-based Laser Ablation Aerosol Mass Spectrometer**

Single particle analysis was conducted using the Aircraft-based Laser Ablation Aerosol Mass Spectrometer (ALABAMA). A detailed description of the instrument can be found in Brands et al. (2011). Briefly, the ALABAMA samples particles through a pressure-controlled inlet and an aerodynamic





lens. The particles are detected and sized by light scattering when passing two continuous laser
beams. Particle ablation and ionization is done by a single 266 nm laser pulse, and the resulting ions
are detected in a bipolar time-of-flight mass spectrometer. Optical detection of aerosol limits the
minimum detectable particle size to approximately 150 nm, and transmission efficiency in the aero-
dynamic lens limits the maximum detectable size to approximately 1000 nm. Particle mass spectra
collected by the ALABAMA are analysed using a software package that includes *m/z* calibration,
peak area integration and automated clustering using fuzzy c-means clustering (Hinz et al., 1999;
Roth et al., 2016). As is done in this case study, subsets of particles can also be analysed manually
by searching for selected marker peaks known from reference laboratory and field data.

A subset of particles 68 particles detected over the period relevant to this case study was analysed
manually using marker peaks as follows. Organic carbon (OC) was characterized by peaks at m/z
27, 37 and 43 ($C_2H_3^+$, $C_3H^+$ and $CH_3CO^+$ or $C_3H_7^+$). Pronounced peaks at m/z multiples of 12
(e.g., 12, 24, ..., 108) ($C_n^{+/-}$) identify elemental carbon (EC). Mass spectra containing peaks at
m/z multiples of 12, but not higher than 36 can be either fragments of elemental or organic carbon
and are therefore designated here as EC/OC. Methanesulfonic acid (MSA) was identified by a peak
at m/z 95 ($CH_3SO_3^-$). Interference from $Na^{37}Cl_2^-$ is unlikely if no m/z 93 ($Na^{35}Cl_2^+$) is present.
Further marker peaks include m/z 97 ($HSO_4^-$) for sulfate (S), m/z 26 and 42 for $CN^-$ and $CNO^-$
(CN), m/z 39 and 41 ($K^+$) for potassium (K), and m/z 40, 56 and 57 ($Ca^+$, $CaO^+$, $CaOH^+$) for
calcium (Ca). The presence of sodium chloride (NaCl) was determined by peaks at m/z 23, 35, 37,
81 and 83 ($Na^+$, $Cl^-$ and $Na_2Cl^+$). Due to chemical aging processes, $Cl^-$ can be replaced by nitrate
resulting in the presence of peaks at m/z 46 and 62 ($NO_2^-$ and $NO_3^-$) in addition to sodium chloride.
Trimethylamine (TMA) was identified by peaks at m/z 58 and 59 ($C_3H_8N^+$ and $C_3H_9N^+$) based
on laboratory reference measurements of TMA particles and previously published field data (e.g.,
Rehbein et al., 2011; Healy et al., 2015).

### 2.7 Identifying airmass history using FLEXPART-WRF

The Lagrangian particle dispersion model *FLEXible PARTicle dispersion model driven by meteorol-
ogy from the* Weather Research and Forecasting model (FLEXPART-WRF) (Brioude et al., 2013)
(website: flexpart.eu/wiki/FpLimitedareaWrf) was used to study the history air masses prior to sam-
pling during the flight. FLEXPART-WRF is based on FLEXPART (Stohl et al., 2005), but uses the
limited area meteorological forecast from WRF (Skamarock et al., 2001), with the specific WRF
forecast details for the NETCARE campaign provided in Wentworth et al. (2015). Here, we use
FLEXPART-WRF run in backward mode to study the origin of air influencing aircraft-based aerosol
measurements. Further details of the FLEXPART-WRF simulations performed for NETCARE 2014
summer campaign are also found in Wentworth et al. (2015).



## 3 Results and Discussion

### 3.1 Flight overview and meteorological situation

In this case study we focus on the flight conducted on 12 July 2014 where Polar 6 travelled at $\sim 3$ km altitude from Resolute Bay, past the marginal ice zone and out over open water to the eastern end of Lancaster Sound (Figure 1a) as far as was permitted by our aircraft range, at which point it descended and returned west. The relevant portion of this flight, over open water in Lancaster Sound, is highlighted in Figure 1a. During this time the aircraft flew to the west at below 100 m a.g.l. and

covered a distance of approximately 200 km under clear sky conditions. Profiles were carried out at three different locations to characterize the vertical structure of the troposphere (Figure 1a, triangles): one profile from 60 m to 3000 m near Resolute Bay ($\sim 95°$ W, denoted as "west" in Figure 1b and c); a second, shallower profile to $\sim 900$ m near the marginal ice zone ($\sim 88°$ W, "central"); and a third profile down from 3000 m to 60 m in eastern Lancaster Sound ($\sim 80°$ W, "east").

Meteorological observations and measurements of trace gases and black carbon indicate a stable and clean atmosphere. Temperature profiles in all three locations indicated a shallow surface-based temperature inversion of $2 - 4$ °C, reaching up to $\sim 800$ m over the ice near Resolute Bay, and to only $\sim 100$ m in the eastern profile (Figure 1b). Applying the method of bulk Richardson number (Aliabadi et al., 2016a) with Polar 6 meteorological observations, and radiosondes conducted

concurrently at Resolute Bay and aboard the CCGS Amundsen, Aliabadi et al. (2016b) estimated boundary layer heights of $275 \pm 164$ m during the NETCARE summer campaign. Here, we will refer to the portion of the boundary layer with a positive vertical gradient in the temperature profile as the "lower boundary layer." Both within and above the lower boundary layer winds were predominantly from the west, with measured wind speeds near the surface averaging 6.5 m s$^{-1}$.

Similarly, surface winds from WRF indicate predominately west-north-west winds at this time, with wind speeds of $4 - 8$ m s$^{-1}$ (Figure S4 in the Supplement). CO profiles in all three locations demonstrated very clean background conditions with concentrations ranging from 73 to 78 ppb$_v$, and little variation with altitude (Figure 1c). Relative humidity was generally high near the surface, with an average of 80% in the lower boundary layer (Figure S5 in the Supplement). Refractory black carbon

(rBC) concentrations also indicate a very clean atmosphere, well below the thresholds for a clean marine boundary layer discussed by Gantt and Meskhidze (2013). Average ($\pm$ standard deviation) rBC mass loadings during the period of interest and over the entire flight were $1.6 \pm 0.9$ ng m$^{-3}$ and $2.5 \pm 1.5$ ng m$^{-3}$, respectively, with slightly higher concentrations found aloft.

   Air mass history from FLEXPART-WRF indicates a strong local Arctic influence on the sampled

air mass. FLEXPART-WRF air mass origin is shown as the column integrated air mass residence time prior to sampling, also referred to as the column integrated potential emission sensitivity (PES), for the release time and location of this case study (Figure 2a). The column integrated PES supports that the locally-influenced air mass originated from generally clean conditions with no pollution sources.



The air mass encountered by the aircraft at 82.2° W and ∼85 m a.g.l. resided over Devon Island for
approximately one week before descending into Lancaster Sound within one day of sampling (Figure
2). The model also indicates that the sampled air mass had a residence time within the lowest 300 m
of four to five hours prior to sampling, providing at least four hours of transport and chemistry within
the boundary layer (Figure 2b and c). Overall, FLEXPART-WRF air mass history suggests that the
sampled air mass had little exposure to fresh sea emissions until four to five hours prior to sampling
when it was exposed to the ocean surface within the lower boundary layer.

### 3.2 Observations of particle growth

Our observations of particle number concentration, over the size range from 5 nm to 1 μm, indicated
the simultaneous presence of nucleation mode and Aitken mode particles near the ocean surface. At
low altitude near 85° W we observed an enhancement in $N_{5-20}$ above background levels, indicating
the presence on nucleation mode particles (Figure 3a). Upon entering the lower boundary layer
further downwind (Figure 1d) and to the east (82.5 – 81° W), we observed a sharp increase in $N_{5-20}$
concurrently with an increase in $N_{20-100}$ (Figure 3a). Particle number size distributions illustrate
that particles below 20 nm (Figure 3b) grow to form a mode centred at 30 – 40 nm (Figure 3c–e).
Beyond 86° W we observe $N_{20-100}$ at background levels of ∼100 cm$^{-3}$. We do not directly observe
the formation of the smallest particles; however, we hypothesize that they were formed through
nucleation in a very clean atmosphere.

The boundary layer was characterized by a low pre-existing aerosol surface area (i.e., a small con-
densation sink). A small number of particles above 200 nm in diameter (∼10 – 15 cm$^{-3}$) were
present within the lower boundary layer, and show a time variation distinct from that of $N_{>50}$
and $N_{>100}$ (Figure 4a). These larger particles are present during both sampling periods within the
lower boundary layer (i.e., near 85° W and 82.5° W), where winds speeds were relatively constant
(6.5±1.8 m s$^{-1}$), and could be from ejection of primary sea-spray aerosol (see below). The small
$N_{>200}$ provides a low pre-existing aerosol surface area (average ± standard deviation: 3.8±2.0 μm$^2$ cm$^{-3}$),
which assists particle nucleation.

$N_{>50}$ and $N_{>100}$ show a time variation distinct from that of the nucleation mode ($N_{5-20}$) and
larger accumulation mode particles ($N_{>200}$). In our eastern-most observations within the lower
boundary layer (82.5 – 81° W) $N_{>50}$, which is accounted for largely by $N_{50-150}$, is enhanced above
background levels of ∼200 cm$^{-3}$ to ∼400 cm$^{-3}$ (Figure 4a). At the same time, CCN concentrations
are elevated to >200 cm$^{-3}$, above background levels of ∼100 cm$^{-3}$ (Figure 4a). $N_{>50}$, $N_{>100}$ and
CCN concentrations remain somewhat elevated up to ∼900 m (Figure 4a, near 80.5° W ), suggest-
ing that some mixing above the lower boundary layer is taking place to the east (profiles of aerosol
number and composition are presented in Figure S5 in the Supplement).

The variation in the size distribution from west to east in Lancaster Sound suggests that particles
between ∼30 nm to greater than ∼50 nm are forming through secondary processes. In our western-



most observations in the lower boundary layer (85.6 – 84.4° W) the size distribution is dominated by
particles below 20 nm (Figure 3a,b). Further to the east in the lower boundary layer (82.5 – 81° W)
$N_{5-20}$ is high and subsequently decreases moving east (Figure 3a), while $N_{>50}$ and $N_{>100}$ do not
increase until 81.9° W (Figure 4a). The aircraft covered a distance of 19 km between entering the
lower boundary layer (82.5° W) and observing this increase in $N_{>50}$ and $N_{>100}$. With a wind speed

of 6.5 m s$^{-1}$ near the surface, the flushing-time over this distance is approximately 50 minutes.
If the aerosol size distribution was dominated by primary sea-spray aerosol, given constant wind
speed, there would be no reason for such a delay in our observations of $N_{>50}$ and $N_{>100}$. Indeed,
given the decreasing abundance of $N_{>300}$, the evidence suggests that the sea spray source, which
is associated with larger particles (see below), is becoming less important as $N_{>50}$ is increasing.

These observations are suggestive of a secondary process growing particles from less than 20 nm
into larger sizes, above 50 nm.

### 3.3 Aerosol composition

#### 3.3.1 Carbonaceous aerosol

Our observations of particle growth are correlated with an increase in organic aerosol (OA) and

methanesulfonic acid (MSA) in sub-micron particles (Figure 4b), corresponding to increased organic-
to-sulfate and MSA-to-sulfate ratios (Figure 5). The presence of MSA, an intermediate-volatility
oxidation product of dimethylsulphide (DMS), indicates a marine-biogenic influence on the aerosol
sulfur (Bates et al., 1992). MSA cannot be viewed as a conservative tracer of DMS oxidation (e.g.,
Bates et al., 1992); however, it is notable that the MSA-to-sulfate ratio reached a peak value of 0.15

during the growth event (corresponding to a peak mass of 60 ng m$^{-3}$), which is significantly higher
than at all other times during this flight (Figure 5). The absolute MSA concentration measured by
the HR-ToF-AMS should be viewed as a lower limit since a portion of the MSA mass could reside
on particles smaller than the lower size-limit of the instrument. Particle-size-resolved mass spectra
(pToF, Figure S6 in the Supplement) during particle growth indicate that total organic aerosol was

present in relatively small particle sizes, from less than 80 nm to approximately 200 nm (vacuum
aerodynamic diameter, $d_{va}$). Unfortunately, signal-to-noise ratios for MSA were such that little use-
ful information could be drawn from the corresponding pToF data. The correlation of OA and MSA
with particle growth suggests that the growth of particles into the size range of the HR-ToF-AMS
was mediated by the condensation of MSA and condensible organic species. The source and iden-

tity of these species, aside from MSA, is not known, but we hypothesize a role for marine-derived
biogenic volatile organic compounds (VOCs).

While non-marine sources of condensible organic species, such as emissions of isoprene and ter-
penes from high Arctic terrestrial vegetation (Schollert et al., 2014) and photochemical production
of VOCs in the snowpack (Grannas et al., 2007), may also contribute to particle growth, single





particle observations of aerosol composition further suggest a marine influence on particles greater
than ∼150 nm ($d_{va}$). Fifty-four percent of particles detected by the ALABAMA over the region
highlighted in Figure 1a contained detectable signal for trimethylamine (TMA, Figure 6), in sup-
port of aerosol growth through the condensation of marine-derived biogenic VOCs (e.g., Facchini
et al., 2008a; Dall'Osto et al., 2012). Consistent with HR-ToF-AMS observations of MSA during
the growth event, ∼30% of particles detected by the ALABAMA contained observable MSA signal.
TMA was mainly present as an internal mixture with potassium, sulfate, other organic species and
to a lesser degree with MSA (Figure 6).

Organic aerosol observed by the HR-ToF-AMS during particle growth appears chemically distinct
from the OA observed at other times during this flight, especially compared to that above the lower
boundary layer (OA mass spectra are presented in Figure S7 of the Supplement). Hydrocarbon frag-
ments ($C_xH_y^+$, largely unsaturated) contribute 50% to growth event OA mass spectra, and only 30%
to non-growth event OA. Oxygenated organic fragments ($C_xH_yO_z^+$) contribute 50% to growth event
OA mass spectra, and 70% to non-growth event OA. $C_xH_y^+$ and $C_xH_yO_z^+$ fragments are correlated
during the growth event, suggesting that these less-oxygenated and more-oxygenated species are
arising from a similar source. Average elemental composition also shows notable differences with
oxygen-to-carbon (O:C) and hydrogen-to-carbon (H:C) ratios in the growth event OA of 0.5 and 1.6
while non-growth event OA was significantly more oxygenated with O:C and H:C ratios of 0.78 and
1.2, suggesting less aged OA during the growth event compared to other times.

To gain further insight into the characteristics of the OA observed during the growth event, we
compared our mass spectrum with a number of OA mass spectra obtained with AMS instruments.
The growth event OA compares favourably with marine-like OA observed at Mace Head, Ireland ($R^2$
= 0.75) (Ovadnevaite et al., 2011) as well as with marine OA observed over the Arctic Ocean ($R^2$ =
0.88) (Chang et al., 2011a). OA from the growth event also compares favourably with alpha-pinene
secondary organic aerosol (SOA) generated under low $NO_x$ conditions ($R^2$ = 0.78) (Chhabra et al.,
2011) and with spectra associated with isoprene SOA from a forested site ($R^2$ = 0.85) (Robinson
et al., 2011), but does not compare well with IEPOX SOA ($R^2$ = 0.07) (Bougiatioti et al., 2013).
In conjunction with the presence of MSA during the growth event, the comparisons with previously
observed marine-OA spectra support the hypothesis that we observe a marine-influenced aerosol.
The comparisons with terpene-related OA could also support a marine-influenced aerosol (e.g., Shaw
et al., 2010), but could also be consistent with other regional sources of these OA precursors (e.g.,
Grannas et al., 2007; Schollert et al., 2014).

### 3.3.2  Other aerosol chemical species

Other aerosol components detected by the HR-ToF-AMS showed a time variation distinct from
organic aerosol species. Sulfate mass loading was relatively constant, within the lower boundary
layer (Figure 4b), suggesting that it did not contribute significantly to particle growth during this



event. Owing to the relatively slower oxidation of sulfur dioxide to sulfuric acid, it is feasible that MSA resulting from DMS oxidation could be contributing to particle growth while sulfate salts are not. However, this would be inconsistent with the results of Giamarelou et al. (2016). Similarly to the observed OA, sulfate was present in relatively small particles with a peak in the size distribution

slightly larger than that of OA (Figure S6 in the Supplement). Ammonium concentrations are low and not only mirror the time variation of MSA and OA during growth, but also the smooth decrease in sulfate from west to east suggesting that both organic and inorganic species could be partially neutralized by ammonium. The HR-ToF-AMS estimate of aerosol neutralization (accounting for sulfate, nitrate and MSA) peaks at a value of $\sim$0.6 during particle growth (Figure 5).

Exclusively within the lower boundary layer we observe an increase in iodine signal as $I^+$ (m/z 126.90), while no other iodine-containing peaks were observed above mass spectral noise (Figure 4c). Our observations are potentially consistent with those of Allan et al. (2015), who used similar measurements to highlight the possible role of iodine-oxide species in particle nucleation in Arctic regions. Here, $I^+$ shows a modest correlation not only with $N_{5-20}$ but also with $N_{>200}$, since parti-

cles in both size ranges are confined to the lower boundary layer and their variability in time is largely dictated by the aircraft's position (Figure S8 in the Supplement). Without further information about the chemical form of the iodine we observe, it is difficult to discern whether the HR-ToF-AMS $I^+$ arises from iodine-oxides present in small particles or from biological iodine-containing compounds and iodine-containing salts potentially present in primary sea-spray aerosol (e.g., Murphy et al.,

465    1997).

Primary sea spray aerosol was confined to the lower boundary layer and contributed largely to $N_{>200}$. The HR-ToF-AMS signal for $NaCl^+$, qualitatively indicating the presence of sea salt aerosol, is present in the lower boundary layer (Figure 4c and S5 in the Supplement) and correlates well with $N_{>200}$ and $N_{>300}$ (Figure S8 in the Supplement). As mentioned above, the negative relationship be-

tween $N_{>50}$ and both $N_{>300}$ and $NaCl^+$ near 82.5° W suggests a decreasing importance of primary sea spray at the point where the secondary formation is maximum. Consistent with this observation, single particle measurements from the ALABAMA indicate that NaCl-containing particles were present at larger sizes (i.e., peaking at 400 nm $d_{va}$) and, notably, were externally mixed from other particle types containing TMA (Figure 6).

### 3.4 Cloud condensation nuclei (CCN)

CCN concentrations are elevated above background levels during the growth event, and are well-correlated with the number of particles greater than 80 nm ($N_{>80}$, Figure 7). If the particles contributing to CCN concentrations at this time were only composed of ammonium sulfate, under our experimental conditions (i.e., 0.6% supersaturation), we would expect the CCN-activation diame-

ter to be $\sim$40 nm (Petters and Kreidenweis, 2007). A CCN-activation diameter of approximately 80 nm therefore indicates that a species less hygroscopic than ammonium sulfate is contributing to





the CCN we observe. This is consistent with the elevated OA mass loading we measure when CCN concentrations are high (Figure 7, colour scale), while sulfate was relatively low compared to other time periods (Figure 7, marker size).

Since the aerosol was not actively dried and the supersaturation was held constant in the CCNC, in order to allow for rapid measurements, a calculation of the effective aerosol hygroscopicity parameter ($\kappa$) in this case carries a large uncertainty (Petters and Kreidenweis, 2007). In particular, measured particle diameters may be slightly larger than the corresponding dry diameter. The temperature in the inlet line was $10 - 15\ ^{\circ}$C warmer than the ambient temperature so that the relative

humidity (RH) decreased significantly as the aerosol entered the sampling line (i.e., during the case study period, the ambient RH was 80% at $8 - 10\ ^{\circ}$C, and the RH decreased within the inlet to approximately $< 30$%). Using the measured aerosol composition, we estimate that measured particle diameters are up to 10% larger than the corresponding dry diameter.

Nonetheless, this calculation is still illustrative of the organic aerosol properties in this environ-

ment. If the kappa value of the organic aerosol ($\kappa_{\mathrm{Org}}$) is 0.1, and $\kappa$ for the whole aerosol is calculated based on the HR-ToF-AMS organic and sulfate loadings and the known $\kappa$ for ammonium sulfate, then the resulting dry diameter for activation is $\sim$60 nm. From our measurements, the activation diameter seems to be larger than 60 nm so that $\kappa_{\mathrm{Org}}$ of 0.1 could be regarded as an upper limit. If we overestimate aerosol size by 10%, due to incomplete drying in our sampling line, then our estimated

CCN-activation diameter and the calculated dry diameter for activation become more similar. Overall, this illustrates that the organic aerosol was relatively non-hygroscopic with $\kappa_{\mathrm{Org}} \sim 0.1$. This estimate is within the range of $\kappa_{\mathrm{Org}}$ recently measured in a coastal, marine influenced environment by Yakobi-Hancock et al. (2014).

## 4 Conclusions

In this case study, we present evidence that growth of nucleation mode particles in the summertime Arctic can be mediated by the condensation of methanesulfonic acid (MSA) and condensible organic species. Our observations of particle growth, informed by observations of particle composition, suggest a combination of primary and secondary aerosol across the size distribution. We observe the growth of small particles, less than 20 nm, into sizes above 50 nm, while our measurements sug-

gest that ejection of primary sea-spray aerosol contributes to externally mixed particles larger than 200 nm. The small $N_{>200}$, that are likely from direct emissions of sea-spray, could contain a substantial fraction of organic aerosol (OA). However, the majority of OA mass observed here is best correlated with MSA, $N_{>80}$ (dominated by $N_{80-150}$), and the presence of trimethylamine (TMA) suggesting that this OA is largely secondary in origin. As well, it occurs simultaneously with a pe-

riod of pronounced aerosol growth. Together, this indicates that the cloud condensation nuclei (CCN) we observe are largely controlled by secondary processes.



Very few studies have measured aerosol composition at high time resolution in the summertime Arctic. Even fewer studies have provided evidence for secondary organic aerosol formation in Arctic regions, in part owing to the infrequency of measurements in the remote marine boundary layer.

These results highlight the potential importance of secondary marine organic aerosol formation, and its role in growing nucleation mode particles into CCN-active sizes in the clean summertime Arctic atmosphere. Future measurements of nucleation and Aitken mode particle composition coupled to characterization of gas-phase organic species will greatly improve our understanding of particle formation and growth in remote regions, aiding in our ability to understand resulting aerosol-cloud-

climate interactions.

*Acknowledgements.* The authors thank Kenn Borek Air Ltd., in particular our pilots Kevin Elke and John Bayes, as well as our aircraft maintenance engineer Kevin Riehl. We gratefully acknowledge John Ford and David Heath at the University of Toronto Chemistry machine shop for their work racking the HR-ToF-AMS and other instruments for deployment aboard Polar 6. We are grateful to Katherine Hayden (Environment and

Climate Change Canada, ECCC) for loaning us the pressure controlled inlet used with the HR-ToF-AMS. We thank Jim Hodgson and Lake Central Air Services in Muskoka, Jim Watson (Scale Modelbuilders, Inc.), Julia Binder and Martin Gerhmann (Alfred Wegener Institute, AWI), Mike Harwood and Andrew Elford (ECCC), for their support of the integration of the instrumentation in the aircraft. We thank Carrie Taylor (ECCC), Bob Christensen (UofT), Lukas Kandora, Manuel Sellmann and Jens Herrmann (AWI), Desiree Toom, Sangeeta

Sharma, Dan Veber, Andrew Platt, Anne Marie Macdonald, Ralf Staebler, Maurice Watt (ECCC) and Kathy Law (LATMOS) for their support before and during the study. We thank the Biogeochemistry department of MPIC for providing the CO instrument and Dieter Scharffe for his support during the preparation phase of the campaign. We thank the Nunavut Research Institute and the Nunavut Impact Review Board for licensing the study. Logistical support in Resolute Bay was provided by the Polar Continental Shelf Project (PCSP) of Natural

Resources Canada under PCSP Field Project 218-14, and we are particularly grateful to Tim McCagherty and Jodi MacGregor of the PCSP. Funding for this work was provided by the Natural Sciences and Engineering Research Council of Canada through the NETCARE project of the Climate Change and Atmospheric Research Program, the Alfred Wegener Institute and Environment and Climate Change Canada.





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





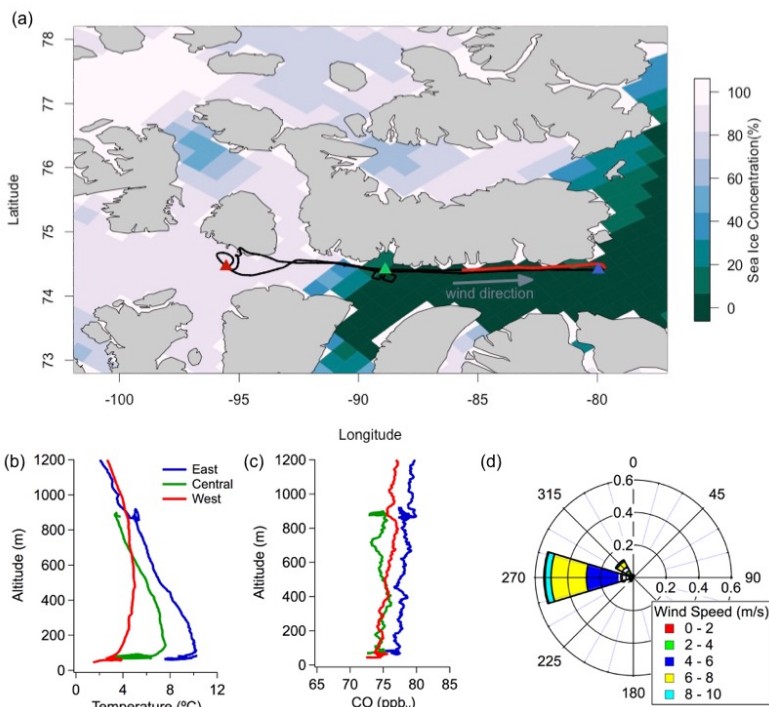

**Figure 1.** (a) Map of the study area showing sea ice concentration for 12 July 2014 from the National Snow and Ice Data Center (nsidc.org, (Cavalieri et al., 1996)) and the flight track originating at Resolute Bay, Nunavut (74° 41' N, 94° 52' W) and extending to eastern Lancaster Sound. The case study area is highlighted in red, at which time the aircraft travelled westward below ∼100 m a.g.l. The prevailing wind direction is marked with an arrow. Triangles mark the location at which the aircraft reached ∼1 km a.g.l during each profile shown in (b) and (c). (b) and (c) Profiles of temperature and CO mixing ratio near Resolute Bay (red), in central Lancaster Sound (green) and in eastern Lancaster Sound (blue). (d) Flight-average wind rose, wind speeds at the surface averaged ∼ 6.5 m s$^{-1}$.





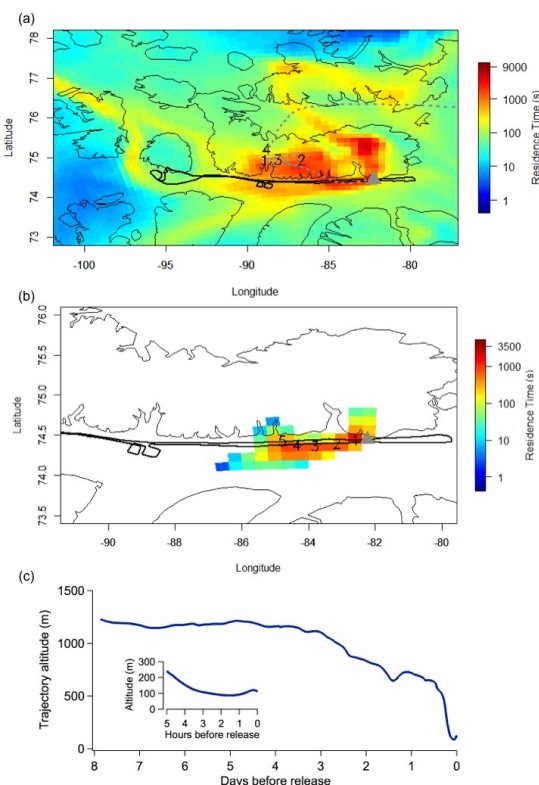

**Figure 2.** (a) Total column airmass residence time predicted by FLEXPART-WRF, indicating the origin of air sampled along the flight track. The location of the aircraft during sampling is noted by the grey triangle, which also indicates the FLEXPART-WRF particle release location (74.4° N, 82.2° W, ~85 m a.g.l., 20:39:25 UTC). The color scale represents the residence time of air, in seconds, at a particular location before arriving at the aircraft position. The plume centroid location is shown with a grey dashed line. Numbers indicate the plume centroid location, in days prior to release. (b) Partial column (below 300 m) PES predicted by FLEXPART-WRF shown as residence time in seconds for particles released at the aircraft location in (a). The color scale shows the residence time particles for five hours prior to the release time and below 300 m. Numbers indicate the plume centroid location, in hours prior to release. (c) Plume centroid altitude eight days prior to release and five hours prior to release (inset).





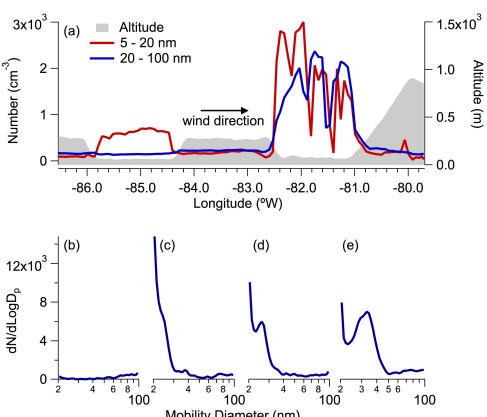

**Figure 3.** (a) Aircraft altitude (grey) and particle number concentrations from 5 – 20 nm ($N_{5-20}$, light blue) and 20 – 100 nm ($N_{20-100}$, dark blue), both shown at the time resolution of the SMS, over the case study area highlighted in Figure 1a. Particle-number size distributions at (b) 85.1 ° W, (c) 82.3° W, (d) 81.8° W and (e) 81.1° W.

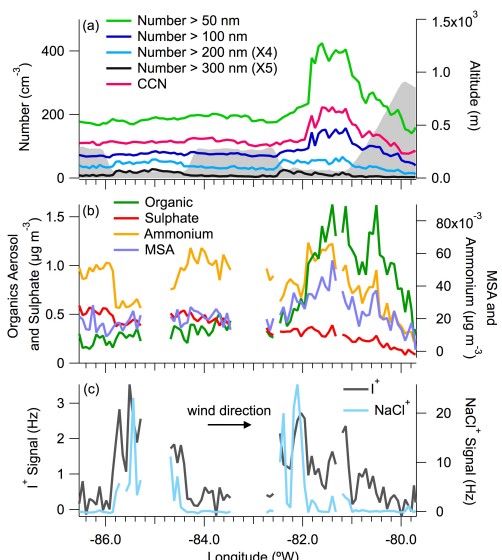

**Figure 4.** (a) Particle number concentrations greater than 50 nm ($N_{>50}$, light green), greater than 100 nm ($N_{>100}$, dark blue), greater than 200 nm ($N_{>200}$, light blue, multiplied by four), greater than 300 nm ($N_{>300}$, black, multiplied by five) and CCN concentrations at 0.6% supersaturation (pink), over the case study area highlighted in Figure 1a. Particle data are shown at the time resolution of the HR-ToF-AMS. (b) Organic species, sulfate, ammonium and methanesulfonic acid (MSA) measured by the HR-ToF-AMS. (c) $I^+$ (m/z 126.90) and $NaCl^+$ signal from the HR-ToF-AMS.



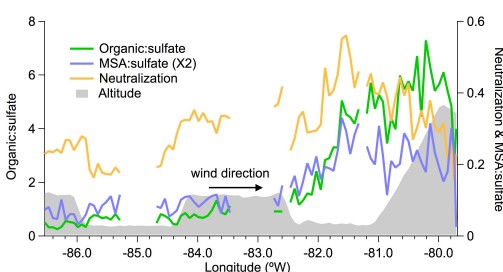

**Figure 5.** Organic-to-sulfate ratio (green), MSA-to-sulfate ratio (purple) and extent of neutralization (orange) over the case study area highlighted in Figure 1a. Altitude is shown in grey on the same scale as Figure 3 and 4. The extent of neutralization is the ratio of measured to predicted ammonium, based on measured sulfate, nitrate and MSA.

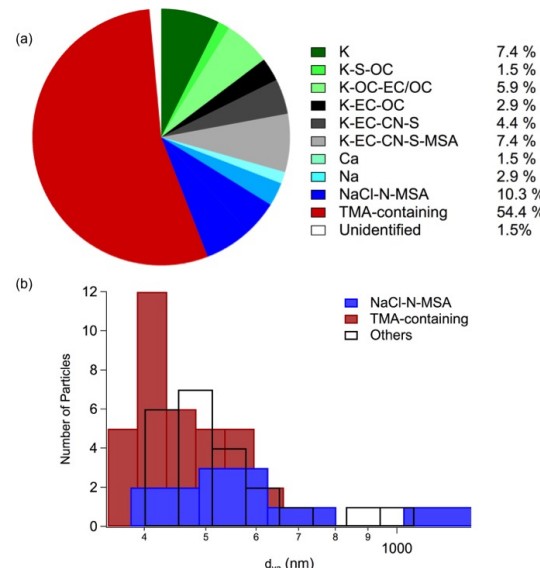

**Figure 6.** (a) Pie chart depicting particle classes detected by the ALABAMA over the case study area highlighted in Figure 1a, with particle class names indicating the relative abundance of the corresponding signals in particle spectra. A total of 68 particle spectra were obtained during the approximately two-hour period; 37 particles contained detectable trimethylamine (TMA) signal and 7 particles contained markers for sea salt (NaCl). TMA-containing particles are mostly internally mixed with K, S, OC and to a lesser degree with MSA and EC/OC. Not all TMA-containing particles included signal for MSA; 13% of all detected particles contained both TMA and MSA signals. (b) Size distributions (in terms of vacuum aerodynamic diameter, $d_{va}$) of TMA-containing particles (red), NaCl-containing particles (blue), and all other particles classes (transparent).





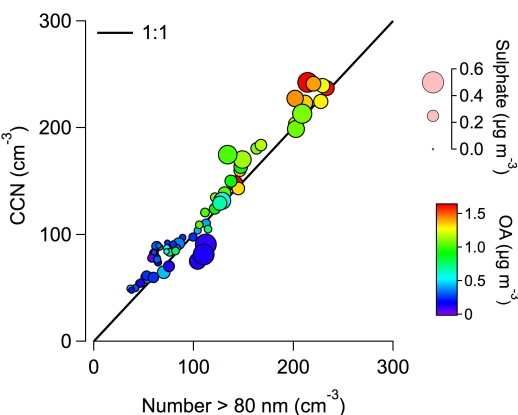

**Figure 7.** Correlation between the number of particles greater than 80 nm ($N_{>80}$, measured by the UHSAS) and the cloud condensation nuclei concentration (CCN) at 0.6% supersaturation, below 1 km, during the case study period. Data are coloured by organic aerosol loading and point size corresponds to sulfate loading.