# Peer review of "Growth of nucleation mode particles in the summertime Arctic: a case study"

_Atmospheric Chemistry and Physics, 2016_

## Referee Comment (RC1) · Anonymous Referee #1 · 12 Apr 2016

This paper describes airborne observations of new particle formation in the Canadian Arctic. Increases in small particle number concentrations were observed during low level flights to the east of Resolute Bay on the edge of the main area of sea ice. As the flight continued in open water the particle size distribution evolved and particles increased in size to 50 nm and greater. As these measurements were taken in very clean conditions, in light winds and when the shallow marine boundary layer was capped with a strong inversion the particle evolution can be linked to new particle formation and growth. Chemical measurements of the larger particles show the presence of methyl sulfonic acid (MSA), organic matter and trimethylamine. The organic matter was shown to have a rather different mass spectral chemical signature during the period of particle growth compared to other regions during the study and points to the role of secondary organic matter in growing new particles into particles that may be active as CCN. There

may be some evidence that iodine is involved in the new particle formation but the authors, rightly, are tentative in their conclusions on this point. The paper is certainly worthy of publication in ACP in my view if some points are considered.

Page 144-165: Given that a comparison of aerosol number concentration across a range of sizes derived from size distributions and total number concentrations is the central theme to the paper I find it strange that a characteristic size distribution from the UHSAS and SMS and the average integrated number comparison with the CPC is not provided as a figure. This could also be used to illustrate how the integrated number concentrations were derived as well as show the agreement between the different instruments the authors refer to. Size distributions up to 100 nm are shown in figure 3 but I suspect that these are only from the SMS.

In the supplementary material (Figure S5), it appears from the profile of Ntotal that there was an initial descent to around 80-100 m and then an ascent to 300 m before the aircraft descended again to 70 m. In this second period of surface layer sampling, Ntotal was not enhanced as it was in the early sampling period. I don't see how this relates to the straight and level runs shown in figure 3 and needs clarifying.

Lines 352 to 353 and Figure 3b: "Particle number size distributions illustrate that particles below 20 nm (Figure 3b) grow to form a mode centred at 30 – 40 nm (Figure 3c–e)." Size distributions in figures 3b, c and d only show size distributions between 20-100 nm. This needs to be explained clearly in the figure caption and text. I don't think that the authors can say that the size distributions on their own show growth from below 20 nm to form a mode at 30-40 nm. I do not dispute the claim but I would like to see a clearer summary of the evidence presented by the authors in this section to support what is at present simply an assertion. This can be done given the distances and timescales. The advection timescale from 85W to 82W is between 3 and 6 hours at windspeeds 4-8 m/s and the sample time of the aircraft is around 10 minutes depending on the aircraft speed. Given the changes in aerosol concentrations and sizes this excludes a wider aerosol source region and implies that the source is to the west

of the sample region and the aerosol distribution develops as the air moves to the east. It would very informative to the reader to include such a discussion at this point in the text in my view and to discount other possibilities.

In addition to the above, can the authors say anything about the growth rates of the particles and the size of the condensation sink?

Lines 368-372: Elevated concentrations of larger particles and aerosol component mass are observed at the east of the sample region at altitudes up to 900 m. The authors suggest that some mixing has occurred. The thermodynamic profiles of potential temperature in figure S5 show an increase with height from the surface to around 300 m, no change to 500 m, and a further increase aloft. This suggest the lowest layer remains stable to 300 m and there is little thermodynamic forcing of mixing throughout the column. Was any cloud present through the column, the RH profile suggest not, but without cloud it is rather difficult to see how mixing of the surface layer could be responsible for the profiles observed.

Minor comments: Line 121: isn't the inner diameter the most important?

Lines 155-157: The authors state that the agreement between different aerosol instruments was generally within a factor of two. What was within a factor of two, total number, size or something else? The comment needs to be more precise.

Line 156: The CCNC is not mentioned at all in the text up to this point. I assume that this is a cloud condensation nucleus counter but the model and operating mode is not mentioned, was it run at one or more supersaturations or was it scanned, if so over what timescale and over what supersaturation range?

Line 269-270: why two laser beams? are these separated or is the sample volume of the two co-located? Page 28: Figure 2 caption: "The location of the aircraft during sampling is noted by the grey triangle" I assume the grey triangle refers to the position of the aircraft at the time of the start of the FLEXPART release?

[Figure]

Line 305 and following: It would be useful to provide some detail on the time of take-off and the air speed and/or time of the profiles and manoeuvres.

Table S1: The ion m/z are incorrect for CH3SO+ and CH3SO2+

Figure S5: caption, should read "below the inversion" and not "in the inversion"

Line 350: "of" not "on"

Page 11-12: Figure 6: Given the very low particle numbers the ALABAMA instrument is limited by counting statistics. It is rather disingenuous to provide the cluster abundances as a fraction when the total numbers in each cluster are less than the fractional amount. It is better to show the total numbers of particles counted. By my calculation, only the TMA containing cluster includes more than 10 particles in the cluster and many of the "clusters" are only 1, 2 or 3 particles.

Figure S6: It is probably best to present the 4 point smoothing and the uncertainty based on the Poisson counting stats which I suspect will show there is little that is statistically significant above 200 nm.

Lines 451-453: I am unconvinced that the ammonium shows a smooth decrease from west to east similar to the sulfate. There is a marked reduction in ammonium in the western boundary layer that is not matched by the sulfate.

---

## Referee Comment (RC2) · Anonymous Referee #2 · 5 May 2016

The manuscript "Growth of nucleation mode particles in the summertime Arctic: a case study" by Willis et al., describes physicochemical properties of atmospheric nanometer-sized particles during a summertime new particle formation event in the Canadian Arctic Archipelago. As the authors correctly point out, new particle formation events, which can form in summer in the Arctic due to clean conditions and higher photochemical activity, are considered to be an important source of cloud condensation nuclei in this region. Because of this, knowledge of the sources and mechanisms of these events are important in order to assess the coupling between terrestrial processes and the atmospheric hydrological cycle. This study makes an important contribution this understanding by providing high quality measurements. They are presently clearly, and in a well-organized manner.

I cannot find many flaws in this study and manuscript; however since it is my job to

provide helpful comments I offer the following suggestions that I hope might improve the overall quality of this manuscript.

1. Since the air mass for this day has spent a week over land (Devon Island), it would be helpful for the reader to know the nature of the land surface and possible sources of condensable gas precursors.

2. Figures 3 and 4. In the text, the authors discuss the lack independent behavior of the number concentrations of particles larger than 50 nm and those between 5 and 20 nm. That difference is best depicted by including the latter on top of the stack of plots in Figure 4. Not such a big deal, but it would allow for closer comparison of the differences between these distributions.

3. Figures 3 b-e show steady growth of the nucleation mode as the aircraft samples downwind. Since "growth" is such a critical aspect of this manuscript (the word appears 52 times in this manuscript), this reader at least is interested in seeing an estimate of the growth rate. This should be feasible given the steady wind conditions and data obtained in this study.

4. Figure 6: correct x-axis to show more clearly the range of particle diameter (it appears that the range starts with sub-10 nm diameters). Also, if the diameter is on the x-axis starts at 300 nm, and the minimum detectable size is 150 nm, then why weren't smaller particles detected by ALABAMA?

Minor editorial comment: For consistency, change the spelling of "sulfate" in Figure 4.

---

## Author Comment (AC1) · 2 Jun 2016

**Response to Anonymous Referee # 1**

We thank Referee # 1 for their comments and suggestions that have helped to improve this manuscript. Our responses to comments and the corresponding changes to the manuscript are detailed below in blue text.

**Major Comments**

This paper describes airborne observations of new particle formation in the Canadian Arctic. Increases in small particle number concentrations were observed during low level flights to the east of Resolute Bay on the edge of the main area of sea ice. As the flight continued in open water the particle size distribution evolved and particles increased in size to 50 nm and greater. As these measurements were taken in very clean conditions, in light winds and when the shallow marine boundary layer was capped with a strong inversion the particle evolution can be linked to new particle formation and growth. Chemical measurements of the larger particles show the presence of methyl sulfonic acid (MSA), organic matter and trimethylamine. The organic matter was shown to have a rather different mass spectral chemical signature during the period of particle growth compared to other regions during the study and points to the role of secondary organic matter in growing new particles into particles that may be active as CCN. There may be some evidence that iodine is involved in the new particle formation but the authors, rightly, are tentative in their conclusions on this point. The paper is certainly worthy of publication in ACP in my view if some points are considered.

**Page 144-165:** Given that a comparison of aerosol number concentration across a range of sizes derived from size distributions and total number concentrations is the central theme to the paper I find it strange that a characteristic size distribution from the UHSAS and SMS and the average integrated number comparison with the CPC is not provided as a figure. This could also be used to illustrate how the integrated number concentrations were derived as well as show the agreement between the different instruments the authors refer to. Size distributions up to 100 nm are shown in figure 3 but I suspect that these are only from the SMS. A characteristic size distribution observed during the case study period has been added as the new Figure 1 in the revised manuscript. Since the UCPC measured particles 5 nm and larger, a comparison of the average integrated concentrations from the UCPC with that of the SMS and UHSAS

yields the number of particles between 5 and 20 nm. Therefore, we do not expect the integrated UCPC concentration to agree well with that of the SMS and UHSAS when small particles are present. The agreement mentioned in section 2.3 was in reference to the SMS and UHSAS in their overlapping size range, which is now illustrated in the new Figure 1 and detailed in the text of section 2.3 (see response to specific comments, below).

**In the supplementary material (Figure S5),** it appears from the profile of Ntotal that there was an initial descent to around 80-100 m and then an ascent to 300 m before the aircraft descended again to 70 m. In this second period of surface layer sampling, Ntotal was not enhanced as it was in the early sampling period. I don't see how this relates to the straight and level runs shown in figure 3 and needs clarifying. We agree that the profile of Ntotal in Figure S5 could be a source of some confusion. Therefore, for consistency with the main text figures, we have replaced the profile of Ntotal with the profile of $N_{5-20}$. It is correct that there was an initial descent into the lower boundary layer, followed by an ascent to approximately 300 m, before we descended into the lower boundary layer again. This can be seen in the altitude traces shown in Figures 3 and 4 (Figures 4 and 5 in the revised manuscript). As is also evident from the traces of $N_{5-20}$ and $N_{20-100}$ in Figure 3, the total number concentration near 85W is much lower than that near 82W.

**Lines 352 to 353 and Figure 3b:** "Particle number size distributions illustrate that particles below 20 nm (Figure 3b) grow to form a mode centred at 30 - 40 nm (Figure 3c–e)." Size distributions in figures 3b, c and d only show size distributions between 20-100 nm. This needs to be explained clearly in the figure caption and text. I don't think that the authors can say that the size distributions on their own show growth from below 20 nm to form a mode at 30-40 nm. I do not dispute the claim but I would like to see a clearer summary of the evidence presented by the authors in this section to support what is at present simply an assertion. This can be done given the distances and timescales. The advection timescale from 85W to 82W is between 3 and 6 hours at

windspeeds 4-8 m/s and the sample time of the aircraft is around 10 minutes depending on the aircraft speed. Given the changes in aerosol concentrations and sizes this excludes a wider aerosol source region and implies that the source is to the west of the sample region and the aerosol distribution develops as the air moves to the east.It would very informative to the reader to include such a discussion at this point in the text in my view and to discount other possibilities. In addition to the above, can the authors say anything about the growth rates of the particles and the size of the condensation sink? We agree that a more clear summary of the evidence for aerosol growth at this point in the text would strengthen this manuscript. To this end, we have expanded the size distributions in Figure 3 (Figure 4 in the revised manuscript) to a range of 20 – 1000 nm by including the UHSAS data. We have also made this clear in the figure caption and text.

We have attempted to estimate the aerosol growth rate from the size distributions observed by the SMS and UHSAS between 82W and 81W (between 86W and 84.4W the number distribution appeared to be dominated by N5-20, with a mode of larger particles of consistent size near 100 nm). Such an estimation is complicated for three main reasons. First, we must estimate the advection time using the average wind speed (6.5 m/s) in the lower boundary layer, neglecting any turbulent motions and potentially underestimating the true transport time. Second, we measured size distributions with SMS (20-100 nm) with a scan time of 60 seconds (40 second up-scan, 20 second down-scan) meaning that we have a total of 10 size distributions over a large spatial area (82W to 81W), and that each size distribution is averaged over a distance of ∼4 km. It is certainly possible for the size distribution to change over 4 km; indeed, we see evidence for this in some size distributions that appear to contain more than one mode below 50 nm. Finally, using these 10 size distributions we must assume that aerosol growth is steady between 82W and 81W, and that the source of condensing material is located at a point to the west of 86W, in order to estimate a growth rate. Under these assumptions, along with the log-normal distribution function method described by Kulmala et al. (2012) and fitting two modes between 20 – 55 nm and 55 –

800 nm, respectively, we find that the mode of smaller particles grew at a rate of 6.6 nm/hr. Additionally, we find the mode of larger particles present between 86–84.4W, peaking at ~85 nm, decreased in size from 82W to 81W. As noted in section 3.2 the lower boundary layer was characterized by a low pre-existing aerosol surface area of $3.8\pm2.0\ \mu m^2\,cm^{-3}$, likely assisting particle formation. The apparent decrease in size of this larger mode is likely because smaller particles were growing into larger sizes giving the appearance of a decrease in the geometric mean diameter above 55 nm. We believe that this relatively large estimated growth rate carries a significant uncertainty for the reasons described above, and therefore have chosen not to include this estimation in the manuscript.

To reflect these points, we have modified the first paragraph (second paragraph in the revised manuscript) of section 3.2 as follows: " Particle number size distributions from 20 – 1000 nm illustrate that particles below 20 nm (Figure 4a,c) grow to form a mode centred at 30 – 40 nm (Figure 4d–f). Beyond 86W we observe $N_{20-100}$ at background levels of ~100 $cm^{-3}$. These observations suggest that the aerosol size distribution develops as the airmass moves downwind to the east. The advection time scale from 85.8 – 81.1W is 6.7 hr, given an average wind speed of 6.5 m/s, and the sampling time of the aircraft over this distance is 35 min. Given the substantial changes in aerosol size and number concentration observed over this relatively short time period, our observations suggest that a source of condensible material contributing to aerosol growth is present to the west of the sampling region and a wider source region is unlikely to contribute to these observed changes. Any estimate of the growth rate in this case is associated with a large uncertainty, since it is complicated by a number of factors including the one-minute time resolution of the SMS that corresponds to a sampling distance of ~4 km, and uncertainties in the advection time. Therefore, it is difficult to quantitatively follow the evolution of the size distribution. Compounded by our lack of knowledge of the spatial uniformity of the condensible material, we do not present an estimate of the growth rate here."

**Lines 368-372:** Elevated concentrations of larger particles and aerosol component mass are observed at the east of the sample region at altitudes up to 900 m. The authors suggest that some mixing has occurred. The thermodynamic profiles of potential temperature in figure S5 show an increase with height from the surface to around 300 m, no change to 500 m, and a further increase aloft. This suggest the lowest layer remains stable to 300 m and there is little thermodynamic forcing of mixing throughout the column. Was any cloud present through the column, the RH profile suggest not, but without cloud it is rather difficult to see how mixing of the surface layer could be responsible for the profiles observed. This flight took place largely under clear sky conditions, and there was no cloud that could have contributed to mixing in this case. FLEXPART-WRF suggests some downslope flow off Devon Island that could have created some mixing in hours prior to our measurements, and wind directions shift slightly as we move above the lower boundary layer, from west towards north-west. This is consistent with the results from WRF. We do agree that the thermodyanmic profiles do not suggest mixing has occurred, and we have altered this discussion as follows: "$N_{>50}$, $N_{>100}$ and CCN concentrations remain somewhat elevated up to ∼900 m (Figure 4b, near 80.5W ), suggesting that some mixing above the lower boundary layer occurred during some time prior to our observations, possibly due to katabatic winds off Devon Island suggested from the Flexpart analyses (Figure 3c). Profiles of aerosol number and composition are presented in Figure S5 in the Supplement."

**Minor Comments**

**Line 121:** isn't the inner diameter the most important? Thank-you for catching this error, this information has been added to the sentence as follows: "Aerosol flowed into the cabin through a stainless steel manifold (outer diameter = 2.5 cm, inner diameter = 2.3 cm) and was directed to the various particle instruments..."
**Lines 155-157:** The authors state that the agreement between different aerosol instruments was generally within a factor of two. What was within a factor of two, total number, size or something else? The comment needs to be more precise. This sentence has been revised to: "Particle number concentrations from the SMS and UHSAS generally agreed within a factor of two over their overlapping size range (70 – 100 nm)."

**Line 156:** The CCNC is not mentioned at all in the text up to this point. I assume that this is a cloud condensation nucleus counter but the model and operating mode is not mentioned, was it run at one or more supersaturations or was it scanned, if so over what timescale and over what supersaturation range? The Cloud Condensation Nuclei Counter (CCNC) is described in section 2.4. This reference to the CCNC in section 2.3 has been removed for clarity. The CCNC was operated at 0.6% supersaturation, as described in section 2.4.

**Line 269-270:** why two laser beams? are these separated or is the sample volume of the two co-located? The two laser beams are separated so that the particle time-of-flight can be measured, and the particle vaccum aerodynamic diameter can be determined. This sentence has been revised to: "The particles are detected and sized by light scattering when passing two continuous laser beams separated along the path of the sampled aerosol."

**Page 28: Figure 2 caption:** "The location of the aircraft during sampling is noted by the grey triangle" I assume the grey triangle refers to the position of the aircraft at the time of the start of the FLEXPART release? Yes, this is also the location of the FLEXPART release shown in Figure 2. This portion of the figure caption has been revised for clarity as follows: "The aircraft location at the time of the FLEXPART-WRF particle release is indicated with a grey triangle ..."

**Line 305 and following:** It would be useful to provide some detail on the time of take-off and the air speed and/or time of the profiles and manoeuvres. We have added information on the time at which the aircraft covered the study area, in the first paragraph of section 3.1, as follows: "The relevant portion of this flight, over open water in Lancaster Sound, is highlighted in Figure 2a (79.7W to 86.5W, 20:00 – 21:20 UTC). During this time the aircraft flew to the west below 100 m a.g.l. and covered a distance of approximately 175 km at a survey speed of 75  m/s under clear sky conditions."

**Table S1:** The ion m/z are incorrect for CH3SO+ and CH3SO2+ This error has been corrected.

**Figure S5:** caption, should read "below the inversion" and not "in the inversion" This has been changed.

**Line 350:** "of" not "on" This error has been corrected.

**Page 11-12:** Figure 6: Given the very low particle numbers the ALABAMA instrument is limited by counting statistics. It is rather disingenuous to provide the cluster abundances as a fraction when the total numbers in each cluster are less than the fractional amount. It is better to show the total numbers of particles counted. By my calculation, only the TMA containing cluster includes more than 10 particles in the cluster and many of the "clusters" are only 1, 2 or 3 particles. The legend in part (a) of this figure has been changed to show the total particle number, and the total number of spectra is now indicated in the legend. For clarity, the reference to "clusters" or "classes" has also been removed from this figure caption since particles were manually grouped based on the presence of marker peaks, as described in section 2.6.3.

**Figure S6:** It is probably best to present the 4 point smoothing and the uncertainty based on the Poisson counting stats which I suspect will show there is little that is statistically significant above 200 nm. We have replaced the 4-point smoothing with the log-normal fits to the size resolved data for total organic aerosol and sulfate during the case study period. These log normal fits show a clear modes at 122 nm and 140 nm for organics and sulfate, respectively. These log-normal fits also show that there is little that is statistically significant above 200 nm.

**Lines 451-453:** I am unconvinced that the ammonium shows a smooth decrease from west to east similar to the sulfate. There is a marked reduction in ammonium in the western boundary layer that is not matched by the sulfate. Our intention here was to highlight the observation that ammonium shows some correlation with both organic and inorganic species. To make this point clearer, this sentence has been revised as follows: "Ammonium concentrations are low and show some correlation with organic and inorganic aerosol species, suggesting that OA, MSA and sulfate could be partially neutralized by ammonium."

**References**

Kulmala, M., Petäjä, T., Nieminen, T., Sipilä, M., Manninen, H. E., Lehtipalo, K., Dal Maso, M., Aalto, P. P., Junninen, H., Paasonen, P., Riipinen, I., Lehtinen, K. E. J., Laaksonen, A., and Kerminen, V.-M.: Measurement of the nucleation of atmospheric aerosol particles, Nat. Protocols, 7, 1651–1667, http://dx.doi.org/10.1038/nprot.2012.091, 2012.

---

## Author Comment (AC2) · 2 Jun 2016

**Response to Anonymous Referee # 2**

We thank Referee # 2 for their helpful comments on this manuscript. Our responses to comments and the corresponding changes to the manuscript are detailed below in blue text.

**General Comments**

The manuscript "Growth of nucleation mode particles in the summertime Arctic: a case study" by Willis et al., describes physicochemical properties of atmospheric nanometersized particles during a summertime new particle formation event in the Canadian Arctic Archipelago. As the authors correctly point out, new particle formation events, which can form in summer in the Arctic due to clean conditions and higher photochemical activity, are considered to be an important source of cloud condensation nuclei in this region. Because of this, knowledge of the sources and mechanisms of these events are important in order to assess the coupling between terrestrial processes and the atmospheric hydrological cycle. This study makes an important contribution this understanding by providing high quality measurements. They are presently clearly, and in a well-organized manner. I cannot find many flaws in this study and manuscript; however since it is my job to provide helpful comments I offer the following suggestions that I hope might improve the overall quality of this manuscript.

1. Since the air mass for this day has spent a week over land (Devon Island), it would be helpful for the reader to know the nature of the land surface and possible sources of condensible gas precursors. According to MODIS land cover data (Friedl et al., 2010), as well as visual observations made during the campaign, Devon Island is covered by snow and ice. As mentioned in section 3.3.1, a photochemical source of volatile organic compounds from snow and ice is a potential source that cannot be discounted. However, our observations of methanesulfonic acid and trimethylamine suggests a significant marine contribution. We have added information on the nature of the land cover on Devon Island to section 3.1 and the relevant figure caption, to illustrate other possible sources of condensible gas precursors.

2. Figures 3 and 4. In the text, the authors discuss the lack independent behavior of the number concentrations of particles larger than 50 nm and those between 5 and 20 nm. That difference is best depicted by including the latter on top of the stack

of plots in Figure 4. Not such a big deal, but it would allow for closer comparison of the differences between these distributions. We agree that rearranging the figures in this way would improve the clarity of our discussion. All particle number traces (previously separated between Figures 3 and 4) are now included in one figure, and consequently all composition information from the HR-ToF-AMS is now included in one figure (previously separated between Figure 4 and 5).

3. Figures 3 b-e show steady growth of the nucleation mode as the aircraft samples downwind. Since "growth" is such a critical aspect of this manuscript (the word appears 52 times in this manuscript), this reader at least is interested in seeing an estimate of the growth rate. This should be feasible given the steady wind conditions and data obtained in this study. We certainly agree that growth in an important aspect of this manuscript. We have attempted to estimate the aerosol growth rate from the size distributions observed by the SMS and UHSAS between 82W and 81W (between 86W and 84.4W the number distribution appeared to be dominated by N5-20, with a mode of larger particles of consistent size near 100 nm). Such an estimation is complicated for three main reasons. First, we must estimate the advection time using the average wind speed (6.5 m/s) in the lower boundary layer, neglecting any turbulent motions and potentially underestimating the true transport time. Second, we measured size distributions with SMS (20-100 nm) with a scan time of 60 seconds (40 second up-scan, 20 second down-scan) meaning that we have a total of 10 size distributions over a large spatial area (82W to 81W), and that each size distribution is averaged over a distance of ~4 km. It is certainly possible for the size distribution to change over 4 km; indeed, we see evidence for this in some size distributions that appear to contain more than one mode below 50 nm. Finally, using these 10 size distributions we must assume that aerosol growth is steady between 82W and 81W, and that the source of condensing material is located at a point to the west of 86W, in order to estimate a growth rate. Under these assumptions, along with the log-normal distribution function method described by Kulmala et al. (2012) and fitting two modes between $20 - 55$ nm and $55 - 800$ nm, respectively, we find that the mode of smaller particles grew at a

rate of 6.6 nm/hr. Additionally, we find the mode of larger particles present between 86–84.4W, peaking at ~85 nm, decreased in size from 82W to 81W. This is likely because smaller particles were growing into larger sizes giving the appearance of a decrease in the geometric mean diameter above 55 nm. We believe that this relatively large estimated growth rate carries a significant uncertainty for the reasons described above, and therefore have chosen not to include this estimation in the manuscript.

4. Figure 6: correct x-axis to show more clearly the range of particle diameter (it appears that the range starts with sub-10 nm diameters). Also, if the diameter is on the x-axis starts at 300 nm, and the minimum detectable size is 150 nm, then why weren't smaller particles detected by ALABAMA? The x-axis of Figure 6 has been corrected to more clearly show its range. Due to limitations of the ALABAMA's optical detection and the transmission efficiency of the aerodynamic lens, the ALABAMA detection efficiency depends on size. Particles at approximately 400 nm are detected most efficiently, and particles smaller than approximately 300 nm have a much lower detection efficiency. Some particles down to 150 nm were detected during this flight. But, during the case study period very few particles between 150 and 300 nm were detected by the ALABAMA. The description of the ALABAMA size range (Section 2.6.3) has been modified for better clarity as follows: "Optical detection of aerosol limits the minimum detectable particle size to approximately 150 nm with particles at approximately 400 nm detected at the highest efficiency. The transmission efficiency in the aerodynamic lens limits the maximum detectable size to approximately 1000 nm."

**Specific Comments**

Minor editorial comment: For consistency, change the spelling of "sulfate" in Figure 4. This error has been corrected.

**References**

Friedl, M., Sulla-Menashe, D., Tan, B., Schneider, A., Ramankutty, N., Sibley, A., and Huang, X.: MODIS Collection 5 global land cover: Algorithm refinements and characterization of new datasets, Collection 5.1 IGBP Land Cover, 2010.

Kulmala, M., Petäjä, T., Nieminen, T., Sipilä, M., Manninen, H. E., Lehtipalo, K., Dal Maso, M., Aalto, P. P., Junninen, H., Paasonen, P., Riipinen, I., Lehtinen, K. E. J., Laaksonen, A., and Kerminen, V.-M.: Measurement of the nucleation of atmospheric aerosol particles, Nat. Protocols, 7, 1651–1667, http://dx.doi.org/10.1038/nprot.2012.091, 2012.